# Metabolomic Profiling in Patients with Heart Failure and Exercise Intolerance: Kynurenine as a Potential Biomarker

**DOI:** 10.3390/cells11101674

**Published:** 2022-05-18

**Authors:** Tarek Bekfani, Mohamed Bekhite, Sophie Neugebauer, Steffen Derlien, Ali Hamadanchi, Jenny Nisser, Marion S. Hilse, Daniela Haase, Tom Kretzschmar, Mei-Fang Wu, Michael Lichtenauer, Michael Kiehntopf, Stephan von Haehling, Peter Schlattmann, Gabriele Lehmann, Marcus Franz, Sven Möbius-Winkler, Christian Schulze

**Affiliations:** 1Department of Internal Medicine I, Division of Cardiology, Angiology and Intensive Medical Care, University Hospital Magdeburg, Otto von Guericke-University, 39120 Magdeburg, Germany; tarek.bekfani@med.ovgu.de; 2Department of Internal Medicine I, Division of Cardiology, Angiology and Intensive Medical Care, University Hospital Jena, Friedrich-Schiller-University, 07743 Jena, Germany; mohamed.el_saied@med.uni-jena.de (M.B.); ali.hamadanchi@med.uni-jena.de (A.H.); mhilse@joho.de (M.S.H.); daniela.haase@med.uni-jena.de (D.H.); tom.kretzschmar2@med.uni-jena.de (T.K.); jasmine.wu@med.uni-jena.de (M.-F.W.); marcus.franz@med.uni-jena.de (M.F.); sven.moebius-winkler@med.uni-jena.de (S.M.-W.); 3Department of Clinical Chemistry and Laboratory Diagnostics, Jena University Hospital, 07743 Jena, Germany; sophie.neugebauer@med.uni-jena.de (S.N.); michael.kienhntopf@med.uni-jena.de (M.K.); 4Institute of Physiotherapy, University Hospital Jena, Friedrich-Schiller-University, 07743 Jena, Germany; steffen.derlien@med.uni-jena.de (S.D.); jenny.nisser@sanivadiagnostics.com (J.N.); 5Clinic of Internal Medicine II, Department of Cardiology, Paracelsus Medical University of Salzburg, 5020 Salzburg, Austria; michael.lichtenauer@med.uni-jena.de; 6Department of Cardiology and Pneumology, University of Göttingen Medical Center, 37075 Göttingen, Germany; stephan.von.haehling@med.uni-goettingen.de; 7German Center for Cardiovascular Research (DZHK), 37075 Göttingen, Germany; 8Institute for Medical Statistics, Computer Science and Data Science (IMSID), Jena University Hospital, 07743 Jena, Germany; peter.schlattmann@med.uni-jena.de; 9Department of Internal Medicine III, Division of Endocrinology, Nephrology and Rheumatology, University Hospital Jena, Friedrich-Schiller-University, 07743 Jena, Germany; gabriele.lehmann@amedes-group.de

**Keywords:** heart failure, kynurenine, metabolomic profile, skeletal muscle endurance

## Abstract

Aims: Metabolic and structural perturbations in skeletal muscle have been found in patients with heart failure (HF) both with preserved (HFpEF) and reduced (HFrEF) ejection fraction in association with reduced muscle endurance (RME). We aimed in the current study to create phenotypes for patients with RME and HFpEF compared to RME HFrEF according to their metabolomic profiles and to test the potential of Kynurenine (Kyn) as a marker for RME. Methods: Altogether, 18 HFrEF, 17 HFpEF, and 20 healthy controls (HC) were prospectively included in the current study. The following tests were performed on all participants: isokinetic muscle function tests, echocardiography, spiroergometry, and varied blood tests. Liquid chromatography tandem mass spectrometry was used to quantify metabolites in serum. Results: Except for aromatic and branched amino acids (AA), patients with HF showed reduced AAs compared to HC. Further perturbations were elevated concentrations of Kyn and acylcarnitines (ACs) in HFpEF and HFrEF patients (*p* < 0.05). While patients with HFpEF and RME presented with reduced concentrations of ACs (long- and medium-chains), those with HFrEF and RME had distorted AAs metabolism (*p* < 0.05). With an area under the curve (AUC) of 0.83, Kyn shows potential as a marker in HF and RME (specificity 70%, sensitivity 83%). In a multiple regression model consisting of short-chain-ACs, spermine, ornithine, glutamate, and Kyn, the latest was an independent predictor for RME (95% CI: −13.01, −3.30, B: −8.2 per 1 µM increase, *p* = 0.001). Conclusions: RME in patients with HFpEF vs. HFrEF proved to have different metabolomic profiles suggesting varied pathophysiology. Kyn might be a promising biomarker for patients with HF and RME.

## 1. Introduction

Heart failure (HF) is accompanied by distinct peripheral skeletal and metabolic abnormalities, end-organ dysfunction, and reduced quality-of-life (QoL) and exercise capacity [1,2,3]. Skeletal muscle dysfunction has been linked to impaired exercise performance in various forms of HF [4,5] and was shown to be associated with distinct metabolites abnormalities [6,7].

Metabolic changes taking place in patients with HF are reflected in various body fluids and tissues. Recently, metabolic and structural perturbations in skeletal muscle have been found in patients with heart failure (HF), both with preserved (HFpEF) and reduced (HFrEF) ejection fraction in association with reduced muscle endurance (RME) [8]. Thus, metabolomic analysis could support defining a distinct fingerprinting of various HF phenotypes and expand our knowledge about pathogenic mechanisms underlying HF and its comorbidities such as sarcopenia or more mild forms such as reduced skeletal muscle function [9].

Metabolomic analysis gives insight into several markers, such as phosphatidylcholines (PCs), amino acids (AAs), acylcarnitines (ACs), biogenic amines, and sphingomyelins (SMs) [10]. Kynurenine (Kyn), a biogenic amine and metabolite of tryptophan (Trp), proved to be a marker for reduced exercise capacity and neurological and psychological disorders like depression in animal models [6,11]. Furthermore, metabolites such as ACs seem to be a surrogate of fatty acid oxidation and are strongly associated with markers of disease severity such as N-terminal pro B-type natriuretic peptide (NT-pro-BNP) in HF [12].

The aim of the current study was to test the potential of metabolites like Kyn as a biomarker for the classification of impaired muscle function in patients with HF. Additionally, we hypothesized that the metabolomic profile of patients with RME and HFpEF is distinguished from those with RME and HFrEF. Accordingly, the metabolomic profiles of the above-mentioned groups were described.

## 2. Methods

### 2.1. Study Population

Between September 2016 and June 2017, healthy controls (HC) from the general population and HF patients from the outpatient clinic at the Jena University Hospital were recruited. According to the inclusion and exclusion criteria, 20 HC, 18 HFrEF, and 17 HFpEF patients were qualified to participate in the current study.

Informed consent was obtained from all participants at recruitment time. The ethic commission of the University Hospital of Jena approved the related protocol, which fulfilled the principles of the Declaration of Helsinki.

The terminology of HFpEF and HFrEF is based on the guidelines of the European Society of Cardiology-HF 2021 [13].

The inclusion and exclusion criteria, as well as the study protocol, were published previously [8].

HC-subjects were free from HF. However, they had some risk factors such as arterial hypertension, atrial fibrillation, and diabetes mellitus.

### 2.2. Muscle Strength and Endurance

The exact methodology and measurements of muscle strength and endurance were reported previously [8].

### 2.3. Serum Analyses

The measurement of serum levels of growth differentiation factor 15 (GDF-15) was performed using enzyme-linked immunosorbent assay (ELISA) kits (DY957, R&D Systems, Minneapolis, MN, USA). Further details of the related process of measurement were described previously [8].

### 2.4. Metabolomic Analysis

Using an API4000 liquid chromatography tandem mass spectrometry (LC-MS/MS) system (AB Sciex, Framingham, MA, USA), an AbsoluteIDQ™ kit p180 (Biocrates Life Science AG, Innsbruck, Austria) was used to quantify different metabolites in serum, such as sphingolipids, glycerophospholipids, biogenic amines, ACs, and AAs, as instructed in the manufacturer’s protocol. To evaluate the calibration curves and assure quality controls, the MetIQ software package (Biocrates Life Science AG, Innsbruck, Austria) was used. Different methods were used to analyze AAs, ACs, glycerophospholipids, and sphingolipids, as previously described [14]. Measured concentrations were exported for statistical analysis. Some metabolites, such as glycerophospholipids, were determined semi-quantitatively.

### 2.5. Statistical Analysis

Normally distributed continuous data are presented as mean ± standard deviation (n ± SD). For variables that are not normally distributed, median and interquartile ranges [25–75%] were reported. Categorical data were presented by percentages. The statistical analysis was performed using the Statistical Package for Social Sciences software (SPSS 26, IBM, Armonk, NY, USA). Pearson’s or Spearman’s simple regression and analysis of variance (ANOVA) were used as appropriate. The multivariate model included those variables with *p* < 0.1 in univariate analyses. A statistical significance was indicated by a two-tailed *p*-value < 0.05 or 0.0167 for post-hoc comparisons.

MetaboAnalyst, a web server at McGill University, Montreal, QC, Canada, was used to create heat maps of the metabolites [15]. Data were normalized for each metabolite as previously described [14]. To calculate the ROC curves, the optimal cutoff value was based on the highest Youden Index value (sensitivity + specificity − 1) [16].

## 3. Results

### 3.1. Baseline Characteristics

Baseline characteristics, exercise capacity, and muscle function measurements among the three groups of this cohort were recently published [8]. A summary of basic characteristics is shown in Table 1. Detailed basic characteristics were previously published [8].

### 3.2. Metabolomic Characteristics

Metabolomic profiles in patients with HFrEF and HFpEF showed no significant differences. However, using partial least squares-discriminant analysis (PLS-DA) and heat maps, clear differences were noticed comparing HFrEF and HFpEF on one hand and HC on the other hand (Figure 1A,B). Patients with HFpEF vs. HC and HFrEF vs. HC showed reduced concentrations of AAs and elevated concentrations of biogenic amines such as Kyn (Figure 1C). Only branched-chain AAs (BCAA) such as valine (Val, Figure 1D) and aromatic AAs such as phenylalanine (Phe) were more elevated in HFpEF and HFrEF than in HC (Appendix A). Furthermore, both patients with HFpEF and HFrEF showed, compared to HC, elevated levels of ACs in serum: medium- (C6–C12) and long-chain ACs (>C12) (Appendix A) and reduced levels of SMs, lysophosphatidylcholines, and PCs (Appendix A, Figure 1).

No difference between HFpEF and HFrEF patients in the analyzed metabolites was observed. After adjusting to sex, data remained unchanged, showing the same relationships among the three groups, as previously mentioned.

### 3.3. Metabolomic Profile in Patients with HF and RME

The mean value of muscle endurance of quadriceps was used to divide the cohort into two groups (above or below the mean). We found that those with RME (<mean value) have a distinct metabolomic profile showing reduced concentrations of different AAs and elevated levels of biogenic amines. The inflammatory biomarker GDF-15 was elevated in this group of patients. Clinically, patients with RME scored worse in the 6-MWT (Table 2).

Furthermore, we analyzed the metabolomic profiles of patients with RME vs. preserved muscle endurance (PME) (i.e., >the mean value) in HFpEF, HFrEF, or HC and focused here only on ACs, AAs, and biogenic amines (Figure 2). We found that the only significant difference observed in HC with RME compared to those with HC and PME was a reduced concentration of C14:1-OH. Patients with HFpEF and RME (Figure 3) showed abnormalities mainly in fatty acid metabolism represented by alterations in ACs: reduced unsaturated-, medium-chain-, long-chain-ACs, and medium-/long-chain ACs. The reduced ratio medium-/long-chain-ACs refer to the impairment in mitochondrial long-chain fatty acid oxidation rates and/or reduced mitochondrial fatty acid uptake. C0/(C16 + C18) as an index of *CPT1B*-activity was reduced in HFpEF patients with RME compared to PME (Table 3 and Figure 2A). Additionally, alanine concentration was also reduced (Table 3 and Figure 3). Kyn was shown for the first time to be elevated in patients with HFpEF and RME (Table 3). There was no difference in age or gender between HFpEF and RME on one hand and HFpEF and PME on the other hand.

Perturbations in HFrEF patients with RME were mainly represented by reduced levels of several AAs, including essential amino acids (Table 4 and Figure 2B). Comparing patients with HFpEF and RME vs. HFrEF and RME showed distinct metabolomic profiles in heat maps and in PLA-DS (Figure 2C).

#### Kynurenine as a Potential Biomarker of RME

We tested the potential of metabolites such as Kyn as biomarkers in detecting impaired muscle endurance and found that the AUC for Kyn was 0.83 (95% CI: 0.72–0.94). With a cut-off of 3.115 µM, Kyn showed 70% specificity and 83% sensitivity in detecting RME (Figure 4A).

In a multiple regression model consisting of short-chain-ACs, spermine, ornithine, glutamate, and Kyn, the latest was an independent predictor for RME (95% CI: −13.01, −3.30, B: −8.2 per 1 µM increase, *p* = 0.001) (Table 5).

Additionally, Kyn showed in a simple regression analysis a strong correlation with GDF-15 (r = 0.7, *p* < 0.0001) (Figure 4B).

## 4. Discussion

Metabolomic analysis is an emerging discipline that has the potential to characterize patients with different diseases, such as HF, according to their metabolomic profiles [17]. We describe in the current study a distinct metabolomic profile in patients with HFrEF and HFpEF compared to HC. Patients with HF showed elevated concentrations of ACs and biogenic amines and reduced concentrations of PCs, SMs, and AAs, except for BCAAs, such as Val, and aromatic AAs, such as Phe. To our best knowledge, the findings of our current study report for the first time distinct metabolomic abnormalities in HFpEF patients with RME (reduced unsaturated-, medium-chain-, long-chain-ACs, and medium-/long-chain ACs, reduced Alanine, and elevated Kyn) compared to HFpEF patients with PME. RME was associated with different metabolomic perturbations in HFpEF (disorders in fatty acid uptake and oxidation and mitochondrial metabolism) than in HFrEF (reduced concentrations of AAs) patients. These findings might focus a special spot of light on the distorted metabolic pathways and the associated pathophysiology of the altered function and structure of the skeletal muscle in patients with HFpEF and HFrEF. As a result, this might have direct therapeutic implications.

One additionally important aspect of our results was showing that Kyn is a very sensitive and specific biomarker in detecting RME (AUC 0.83) in patients with HF.

### 4.1. The Metabolism of AAs in Patients with HF and the Role of Kyn in Detecting Patients with RME

Metabolic derangements of the failing heart have been shown in several studies with the focus on fatty acids and glucose metabolism [18,19]. However, less is known about the alterations of AAs and biogenic amines in patients with HF. AAs are important nutrients and potent signaling molecules to produce cardiac energy. This is especially true in hypoxic conditions and in HF [20]. It was reported that BCAAs, such as Val, and aromatic AAs, such as Phe, are elevated in HF in animal models and have a high predictive value for insulin resistance and Type II diabetes mellitus [20]. Sun et al. showed recently in an animal model that a catabolic defect of BCAAs leads to HF [21]. The accumulation of BCAA in the heart muscle in animal models was associated with insulin resistance [22]. Our current results showing an elevation of Phe and Val and a reduction of several other AAs in the serum of humans with HF compared to HC are in line with the above-mentioned studies performed in animal models.

A further important finding of our current study is showing elevated concentrations of Kyn in the serum of patients with HF (HFrEF and HFpEF) and RME. It has been shown recently that Kyn is elevated in patients with HFrEF [11]. We confirmed these results and extended this knowledge to patients with HFpEF. Furthermore, in a multiple regression model consisting of short-chain-ACs, spermine, ornithine, glutamate, and Kyn, the latest was an independent predictor for RME. Additionally, we found a strong association between biogenic amines such as Kyn and inflammatory factors like GDF-15.

Trp-metabolism in physiologic conditions takes place in liver cells using tryptophan 2,3-dioxygenase [23]. However, during inflammatory processes or oxidative stress, such as in HF, Trp is metabolized by indoleamine-pyrrole 2,3-dioxygenase (IDO) in other cell types, mainly in the blood and lymphoid tissues [24]. In the latest conditions, several products, such as Kyn, 3-hydroxykynurenine, anthranilic acid, 3-hydroxyanthranilic acid, kynurenic acid, and quinolinic acid, result from the degradation of Trp. [7]. IDO-related pathways and the resulting elevated Kyn were proven to be involved in cardiometabolic diseases such as metabolic syndrome or atherosclerosis [25,26]. Recently, Kyn was found to lead to cardiomyocyte apoptosis after myocardial infarct in a mice model through the production of reactive oxygen species [27]. Parallel to that, a recent study showed that the distorted Trp-metabolism pathway and the resulting elevated Kyn concentrations were associated with atherosclerosis and myocardial infarct [25]. Our findings are in line with the above-mentioned studies, similarly showing an association between the elevated Kyn concentrations in serum and the impaired function of the skeletal muscle (RME) in patients with HF. This might be suggestive of a generalized Kyn-related pathophysiology in inflammatory processes such as HF that seems to take place in several tissues, such as heart muscle and skeletal muscle. Our results need to be further investigated in prospective studies or in animal models with HF and RME to prove the causality.

### 4.2. Fatty Acid Metabolism in Patients with HF

Fatty acid oxidation perturbation is well-known in patients with HF. The most studied metabolites of this distorted pathway in HF are ACs [28,29,30]. We found in the current study elevated levels of long- and medium-chain ACs in the serum of patients with HFrEF and HFpEF compared to HCs. This is in line with the reduced gene expressions of different fatty acid oxidation genes and proteins in the skeletal muscle, as has been shown recently by our working group [8].

### 4.3. Metabolomic Profile of Patients with HF and RME

We described in our current study the metabolomic profiles of patients with HF and RME. These patients had reduced concentrations of AAs and elevated concentrations of Kyn and GDF-15. Clinically, this was associated with reduced 6-MWT.

We described additionally the metabolomic profiles of patients with RME vs. PME in HFpEF, HFrEF, and HC, respectively, to figure out possible distorted pathways or abnormal metabolomic disciplines. One of the most novel findings of our current study is showing several indices of perturbations of fatty acid oxidation in multiple levels in patients with HFpEF and RME, such as a reduction of medium-chain and long-chain ACs and a decrease in the ratio of medium-/long-chain-ACs. Contrary to what is expected, studies in obese patients with diabetes and in those on high-fat feeding have shown increased rates of β-oxidation in skeletal muscle. This was mainly explained by “incomplete” fat oxidation, where fatty acids enter the mitochondria but are not completely degraded [28]. However, such an analysis has not been performed yet considering the skeletal muscle function of patients with HFpEF and RME. Though our analysis was not performed directly in skeletal muscle tissues, we described for the first time the metabolomic profile of HF patients with RME.

Furthermore, activated β-oxidative genes do not always lead to increased expression of downstream metabolic pathways such as the tricarboxylic acid cycle (TCA cycle), also known as the Krebs cycle, and the electron transport chain. Instead, in some cases, this was associated with decreased expression levels of *PGC-1a*, a transcriptional coactivator that is responsible for mitochondrial biogenesis and function [31]. This all is in line with our above-mentioned results and with the reduced levels of C0/(C16 + C18) in patients with HFpEF and RME, which was proven to be a surrogate of increased activity of *CPT1B* [32]. This all supports the mitochondrial dysfunction in HF with RME and shows that ACs apparently enter the mitochondria but are likely not appropriately metabolized or are “incompletely oxidized” in the mitochondria. This is in line with our previous results showing reduced levels of *MFN-2* in skeletal muscle of patients with HF and RME [8]. Our results need to be confirmed in further and larger prospective studies.

*CPT1B* intermediates in the most limiting-rate step of fatty oxidation: the conversion of fatty acyl-CoA esters into fatty acylcarnitine derivatives, which enter the mitochondria where *CPTB2* convert them back into fatty acyl-CoA in preparation for further mitochondrial β-oxidation [33]. Defects of the β-oxidation that suppresses the oxidation of long-chain fatty acids lead to the accumulation of mitochondrial long-chain acyl-CoAs, which, in turn, makes the mitochondria ineffective. On the other hand, impaired long-chain fatty acid β-oxidation produces decreases in short- and medium-chain acyl-CoAs and finally reduced concentrations of short- and medium-chain AC derivatives as a result of abnormal downstream cycles of long fatty acids of β-oxidation. Therefore, reductions in short-, medium-, and medium-/long-chain AC ratios in the circulation might be a manifestation of impairments in mitochondrial long-chain fatty acid oxidation rates. If fatty acid oxidation is weakened according to inhibition of mitochondrial fatty acid uptake, decreases in short-, medium-, and long-chain ACs could be observed [33]. Our findings show in HFpEF patients and RME an increased fatty acid oxidation rate and, at the same time, give some hints about mitochondrial dysfunction likely in other proteins than *CPT1B*. The causality between RME and fatty acid metabolism needs to be further clarified in future research.

A further abnormality was noted in HFpEF patients with RME that might enhance the reduced energy production in these patients. Alanine concentration was significantly more reduced compared to HFpEF patients with PME. Alanine is exported from skeletal muscles to the liver to support hepatic gluconeogenesis [33].

Patients with HFrEF and RME compared to HFrEF and PME showed mainly reduced concentrations of AAs without any apparent alterations in fatty acid oxidation.

## 5. Limitations

Our results need to be confirmed in further larger prospective studies. The causality should be investigated in cell lines and animal experiments. Ideally, metabolomic analysis should be performed in skeletal muscle samples and in peripheral blood simultaneously. Additionally, because of the small sample size, we tested in the multivariate analysis only a model consisting of different metabolites. However, testing other models, including age and other comorbidities, such as atrial fibrillation, would be reasonable in larger cohorts. Furthermore, previous studies showed that Trp-Kyn metabolites are associated with HF and, to a lesser extent, with atrial fibrillation. The causality between Kyn and atrial fibrillation should be investigated in larger future studies. This is especially true in patients with HF, where atrial fibrillation is a very common comorbidity.

## 6. Conclusions

The metabolic profile of patients with RME among HFpEF and HFrEF patients was shown to be different among these groups of patients and is likely due to different distorted metabolic pathways. Furthermore, our results present a clear association between metabolic and functional changes occurring in skeletal muscle and the metabolomic profiles found in the serum of patients with HF. Additionally, we showed for the first time in HFpEF patients with RME elevated levels of Kyn. Kyn correlated with elevated inflammatory biomarker levels and was an independent predictive factor for RME. Our results will likely give a new direction for future research by focusing on the metabolomic perturbations taking place in patients with HF, especially those with HFpEF and RME. This might finally have direct therapeutic implications.

## Figures and Tables

**Figure 1 cells-11-01674-f001:**
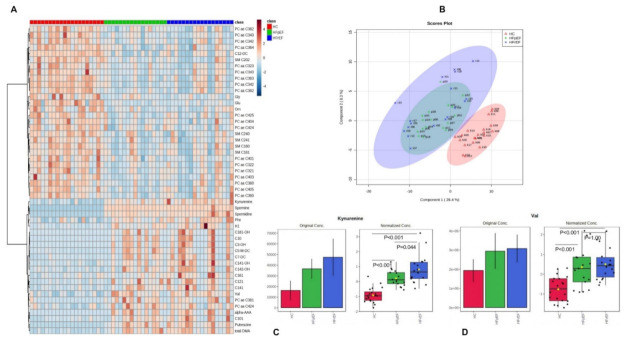
Metabolomic profiles of patients with HFpEF, HFrEF, and HC. (**A**) Visualization of the metabolomic profile using a heat map to show changes in top 50 metabolite concentrations among patients with HFpEF (green), HFrEF (blue), and HC (red). (**B**) Partial least squares-discriminant analysis (PLS-DA) to demonstrate the metabolomic difference among patients with HFpEF (green), HFrEF (blue), and HC (red). (**C**) Concentration of Kynurenine (µM) in serum in patients with HC vs. HFpEF vs. HFrEF. (**D**) Concentration of Valine (µM) in serum in patients with HC vs. HFpEF vs. HFrEF. a: acyl, aa: diacyl, ae: acyl-alkyl, AC: acylcarnitine, Cx:y: where x is the number of carbons in the fatty acid side chain; y is the number of double bonds in the fatty acid side chain, DC: decarboxyl, Glu: Glutamic acid, Gly: Glycine, H1: Sum of Hexoses (90% glucose), OH: hydroxyl, Orn: Ornithine, PC: phosphatidylcholine, Phe: Phenylalanine, SM: sphingomyelin, total DMA: Total dimethylarginine, Val: Valine.

**Figure 2 cells-11-01674-f002:**
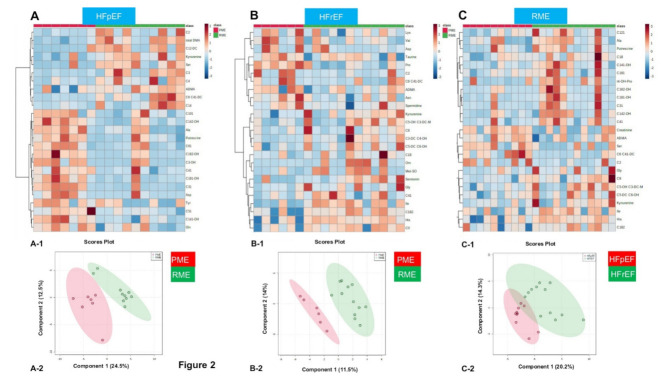
Metabolomic profiles of patients with reduced muscle endurance (RME) vs. preserved muscle endurance (PME) in HFpEF vs. HFrEF suggest different distorted metabolites and likely different mechanisms and metabolic pathways accompanying the RME. Visualization of the metabolomic profile using heat maps to show changes in the top 25 metabolite concentrations, and partial least squares-discriminant analysis (PLS-DA) among HFpEF ((**A**), **A****-1** and **A****-2**), HFrEF ((**B**), **B****-1** and **B****-2**), and in patients with RME and HFpEF vs. RME and HFrEF ((**C**), **C****-1** and **C-2**), respectively.

**Figure 3 cells-11-01674-f003:**
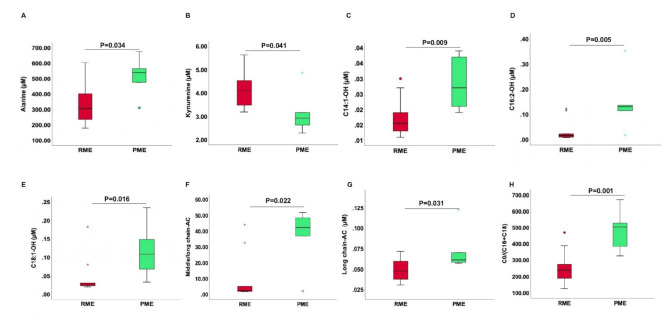
Concentration of different metabolites in serum in patients with HFpEF and RME. The concentration of Alanine (**A**), kynurenine (**B**), C14:1-OH (**C**), C16:2-OH (**D**), C18:1-OH (**E**), medium-/long-chain-ACs (**F**), long-chain-ACs (**G**), and C0/(C16 + C18) (**H**) as an indicator of the function of CPT1B in serum are shown as boxplots in patients with RME (red) vs. PME (green). AC: acylcarnitine, Cx:y: where x is the number of carbons in the fatty acid side chain; y is the number of double bonds in the fatty acid side chain; OH: hydroxyl. PME: Preserved Muscle Endurance; RME: Reduced Muscle Endurance.

**Figure 4 cells-11-01674-f004:**
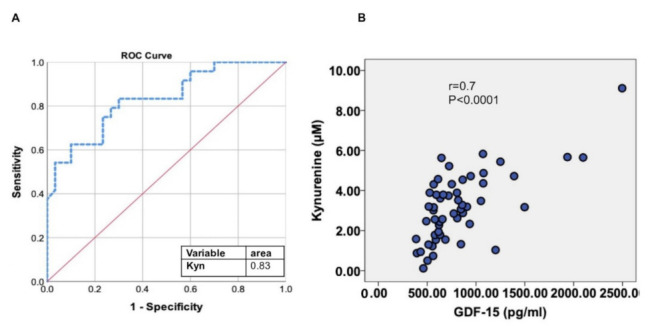
The relationship of Kyn to inflammatory biomarker (GDF-15) and its role in detecting RME. (**A**) The receiver operator characteristic (ROC) curve of Kynurenine to distinguish patients with RME of the left leg in flexion/muscle mass of the left leg. Area under the curve (AUC) is 0.83. (**B**) The relationship between GDF-15 and Kyn is shown in a simple regression (correlation) analysis.

**Table 1 cells-11-01674-t001:** Basic characteristics in patients with HFpEF, HFrEF, and HC.

	HFpEF N = 17	HFrEF N = 18	HC N = 20	*p*-Value
Age (years)	71 ± 6	68 ± 9	66 ± 7	0.1
Sex (m/f) f%	8/9 (53%) ^‡^	15/3 (17%) ^†^	7/13 (65%)	0.0001
BMI (kg/m^2^)	28.7 ± 4.6	27.9 ± 5.3	26.4 ± 4.2	0.2
NYHA (II/III) %	(76.5/23.5) ∗	(83.3/16.7) ^†^	(0/0)	0.0001
LVEF (%)	62.0 [53.0–66.0] ^‡^	30.0 [23.5–32.5] ^†^	61.0 [57.3–66.3]	0.0001
LAVI (ml/m^2^)	34.1 ± 7.1 ∗^,‡^	44.9 ± 19.0^†^	17.2 ± 8.3	<0.0001
E/e’	13.1 [10.6–15.3] ∗^,‡^	15.9 [13.9–24.5] ^†^	9.8 [8.1–11.8]	0.001
BNP (pg/mL)	128 [73–218] ∗^,‡^	317 [181–430] ^†^	35.5 [25.3–56.5]	<0.0001
GFR (mL/min)	72.4 [67.5–82.2]	84.5 [60.1–94.4]	85.3 [70.9–94.3]	0.11
AST (µmol/L)	0.41 [0.38–0.46]	0.45 [0.41–0.57]	0.47 [0.40–0.50]	0.14
ALT (µmol/L)	0.35 [0.30–0.55]	0.43 [0.30–0.57]	0.37 [0.29–0.46]	0.71

∗: *p* < 0.05 in comparison between HFpEF and HC. ^†^: *p* < 0.05 in comparison between HFrEF and HC. ^‡^: *p* < 0.05 in comparison between HFpEF and HFrEF. ALT: Alanine Transaminase, AST: Aspartate Aminotransferase, BMI: Body Mass Index, BNP: Brain Natriuretic Peptide, GFR: Glomerular Filtrating Rate, HC: Healthy Controls, HFpEF: Heart Failure with Preserved Ejection Fraction, HFrEF: Heart Failure with Reduced Ejection Fraction, LAVI: Left Atrial Volume Index, LVEF: Left Ventricle Ejection, NYHA: New York Heart Association.

**Table 2 cells-11-01674-t002:** Metabolomic and clinical profile of patients with HF and reduced vs. normal muscle endurance.

	Endurance of Left Quadriceps ≤ (88.99 Nm/kg) Mean Value N = 29	Endurance of Left Quadriceps > (88.99 Nm/kg) Mean Value N = 25	*p*-Value
Age (years)	70 ± 8	66 ± 7	0.08
Sex (m/f)	15/14	15/10	0.59
Alanine (µM)	431 ± 144	535 ± 139	0.010
Arginine (µM)	66.75 ±	31.06 ±	0.009
Glutamate (µM)	65.20 [52.40–108]	110 [69.55–157]	0.039
Glycine (µM)	197 [137–306]	303 [197–437]	0.09
Ornithine (µM)	109 [87.60–132]	140 [112–178]	0.008
Proline (µM)	272 ± 65.00	307 ± 59.08	0.05
Serine (µM)	94.00 ± 41.13	119 ± 35.18	0.020
Kynurenine/Tryptophan (µM)	0.06 ± 0.03	0.04 ± 0.02	0.001
Kynurenine (µM)	4.01 ± 1.92	2.43 ± 1.27	<0.0001
short-chain ACs (µM)	0.10 [0.08–0.13]	0.08 [0.07–0.1]	0.017
medium-chain ACs (µM)	0.11 [0.08–0.37]	0.10 [0.08–0.66]	0.80
long-chain ACs (µM)	0.05 [0.04–0.06]	0.05 [0.04–0.07]	0.67
GDF-15 (pg/mL)	838 [615–1073]	621 [508–790]	0.005
6-MWT (m)	517 [383–555]	572 [520–614]	0.002

AC: acylcarnitine, GDF-15: growth differentiation factor 15, 6-MWT: 6-min walk test.

**Table 3 cells-11-01674-t003:** Metabolomic profile of patients with HFpEF and RME vs. PME.

	RME (≤74.77 Nm/kg) N = 10	PME (>74.77 Nm/kg) N = 6	*p*-Value
**Age (years)**	72 ± 7	69 ± 7	0.38
**Sex (m/f)**	3/7	5/1	0.12
**Alanine (µM)**	347 ± 151	518 ± 121	0.034
**Asparagine (µM)**	54.84 ± 11.47	42.88 ± 4.91	0.031
**Kynurenine (µM)**	4.10 ± 0.77	3.13 ± 0.91	0.040
**C0 (µM)**	34.70 [31.30–37.43]	39.45 [36.0–67.05]	0.09
**C14:1-OH (µM)**	0.02 ± 0.01	0.03 ± 0.01	0.009
**C14:2-OH (µM)**	0.01 ± 0.01	0.03 ± 0.01	0.002
**C16:2-OH (µM)**	0.02 [0.01–0.05]	0.13 [0.09–0.19]	0.005
**C18:1 (µM)**	0.16 [0.11–0.20]	0.20 [0.20–0.31]	0.011
**C18:1-OH (µM)**	0.03 [0.02–0.04]	0.11 [0.06–0.17]	0.016
**C0/(C16 + C18)**	254 ± 103	486 ± 122	0.001
**Medium-/long-chain ACs**	2.43 [1.75–11.98]	42.30 [28.10–49.28]	0.022
**Medium-chain ACs (µM)**	0.12 [0.08–0.33]	0.75 [0.40–0.81]	0.011
**Long-chain ACs (µM)**	0.05 ± 0.01	0.07 ± 0.03	0.031
**Unsaturated ACs (µM]**	0.07 ± 0.02	0.11 ± 0.02	0.005

**Table 4 cells-11-01674-t004:** Metabolomic profile of patients with HFrEF and RME vs. PME.

	RME (≤87.34 Nm/kg) N = 12	PME (>87.34 Nm/kg) N = 6	*p*-Value
**Age (years)**	71 ± 8	62 ± 5	0.022
**Sex (m/f)**	10/2	5/1	1.00
**Leucine (µM)**	151 ± 34.6	210 ± 77.27	0.036
**Proline (µM)**	268 ± 43.18	346 ± 33.23	0.001
**Threonine (µM)**	101 ± 21.49	134 ± 35.0	0.024
**Kynurenine (µM)**	5.22 ± 1.94	3.74 ± 0.36	0.09
**Valine (µM)**	283 ± 53.47	359 ± 82.16	0.030
**Essential amino acids (µM)**	132 ± 17.93	164 ± 38.01	0.027

AC: acylcarnitine, Cx:y: x is the number of carbons in the fatty acid side chain; y is the number of double bonds in the fatty acid side chain, OH = hydroxyl, PME: Preserved Muscle Endurance, RME: Reduced Muscle Endurance.

**Table 5 cells-11-01674-t005:** Linear regression model with RME as the dependent variable.

Heading	Univariate			Multivariable		
	B	95% CI	*p*-Value	B	95% CI	*p*-Value
**Alanine (µM)**	0.08	0.02, 0.13	0.013	0.21		0.12
**Asparagine (µM)**	0.04	−0.60, 0.67	0.90			
**Glutamate (µM)**	0.18	0.01, 0.34	0.035	0.10		0.53
**Glycine (µM)**	0.03	−0.02, 0.08	0.28			
**Ornithine (µM)**	0.25	0.02, 0.49	0.036	0.11		0.45
**Phenylalanine (µM)**	−0.21	−0.96, 0.53	0.57			
**Tryptophan (µM)**	0.08	−0.52, 0.69	0.78			
**Valine (µM)**	−0.02	−0.12, 0.09	0.78			
**Essential amino acids (µM)**	0.04	−0.24, 0.33	0.76			
**Kynurenine (µM)**	−7.62	−12.31, −2.93	0.002	−8.20	−13.01, −3.30	0.001
**Spermidine (µM)**	−45.77	−189, 97.76	0.53			
**Spermine (µM)**	−133	−262, −4.60	0.043	0.02		0.91
**Short-chain ACs**	−159	−347, 29.25	0.096	-0.07		0.60
**Medium-chain ACs**	0.98	−27.50, 29.46	0.95			
**Long-chain ACs**	152	−263, 567	0.47			

AC: acylcarnitine.

## Data Availability

Appendix A are avialabe.

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
