# Peer review of "Metabolomic Profiling in Patients with Heart Failure and Exercise Intolerance: Kynurenine as a Potential Biomarker"

_cells, 2022, doi:10.3390/cells11101674_

Round 1

Reviewer 1 Report

This is  a cross-sectional study which sought to determine the association of metabolites with RME in HF and healthy control subjects.  Patients with HFpEF and RME showed reduced levels of long-chain-, medium-chain-, medium-/long-chain-ACs ratios. The patients with HFrEF and RME showed reduced concentrations of AAs. In multiple regression, Kyn was an independent predictor for RME. Kyn showed discrimination ability for detecting RME. From these findings the Authors concluded that Kyn has high potential as a biomarker for classifying RME. The manuscript is well written and the findings looks reasonable.

The Reviewer as just several comments on this paper.

Minor comments:

Figures 1 and 2. The figures are too small to see. Please enlarge them so that the readers can see them.

Table 2.Total number of patients was 54. A total of 55 patients were enrolled in this study. How about missing 1 patient?

Figure 4. The Reviewer assumes that this is not regression analysis but correlation analysis.

Author Response

This is  a cross-sectional study which sought to determine the association of metabolites with RME in HF and healthy control subjects.  Patients with HFpEF and RME showed reduced levels of long-chain-, medium-chain-, medium-/long-chain-ACs ratios. The patients with HFrEF and RME showed reduced concentrations of AAs. In multiple regression, Kyn was an independent predictor for RME. Kyn showed discrimination ability for detecting RME. From these findings the Authors concluded that Kyn has high potential as a biomarker for classifying RME. The manuscript is well written and the findings looks reasonable.

The Reviewer as just several comments on this paper.

We would like to thank the reviewer for the valuable feedback and suggestions.

Minor comments:

Figures 1 and 2. The figures are too small to see. Please enlarge them so that the readers can see them.

The figures were optimized as suggested.

Table 2.Total number of patients was 54. A total of 55 patients were enrolled in this study. How about missing 1 patient?

The only missing value is the endurance strength of the left leg in patients. Thus, the resulting number 54 instead of 55.

Figure 4. The Reviewer assumes that this is not regression analysis but correlation analysis.

Figure 4-B refers to simple regression or correlation analysis. We updated this accordingly in the figure legends.

Reviewer 2 Report

Bekfani, Schulze and colleagues in their paper report an elegant analysis on the connection between Kynurenine, amino acid concentrations and the muscle endurance in patients with heart failure. They compared these biomarkers in patients with HFrEF, HFpEF and healthy control. At multivariable analysis Kynurenine showed an independent correlation with exercise tolerance. The methods are quite sound and the paper is easy to read. The field is interesting and with a potential clinical impact.

I have the following concerns.

Major points

1) Heart failure patients were categorised in HFpEF and HFrEF. However, last available guidelines divide patients in HFrEF, HFpER and HFmrEF. In this view, patients should be divided in these three categories. Otherwise, the HFrEF group should be named “HFrEF/HFmrEF”

2) Previous reports showed a significant correlation between atrial fibrillation, heart failure and Kynurenine concentrations (doi: 10.1093/ajcn/nqab238.). On the other hand, atrial fibrillation is a strong prognosticator in heart failure (doi: 10.1016/j.ijcard.2020.08.062). These two aspects should be discussed.

3) Some patients from the healthy control group suffer from comorbidities such as hypertension and AF. Even though this is not a relevant limitation, in the methods section it should be reported that control subjects were only free from heart failure.

Minor points

  • Please put HC in brackets after healthy controls
  • I see that in univariable and multivariable analyses prognostic factors other than biomarkers (i.e. atrial fibrillation, age, sex, etc.) were not tested. This limitation should be acknowledged.
  • Please report the number of missing for each variable.
  • Please refer to latest available heart failure guidelines (doi: 10.1002/ejhf.2333.).
  • References should be formatted according to journal policy.

Author Response

Bekfani, Schulze and colleagues in their paper report an elegant analysis on the connection between Kynurenine, amino acid concentrations and the muscle endurance in patients with heart failure. They compared these biomarkers in patients with HFrEF, HFpEF and healthy control. At multivariable analysis Kynurenine showed an independent correlation with exercise tolerance. The methods are quite sound and the paper is easy to read. The field is interesting and with a potential clinical impact.

I have the following concerns.

We would like to thank the reviewer for the valuable comments and feedback.

Major points

1) Heart failure patients were categorised in HFpEF and HFrEF. However, last available guidelines divide patients in HFrEF, HFpEF and HFmrEF. In this view, patients should be divided in these three categories. Otherwise, the HFrEF group should be named “HFrEF/HFmrEF”.

Thank you for the important remark. We have recruited in the current analysis patients with LVEF < 40% and patients with >50%. None of the patients have the LVEF between 40-50%. Thus, we described only HFrEF and HFpEF. No patients with HFmrEF were included in this study.

2) Previous reports showed a significant correlation between atrial fibrillation, heart failure and Kynurenine concentrations (doi: 10.1093/ajcn/nqab238.). On the other hand, atrial fibrillation is a strong prognosticator in heart failure (doi: 10.1016/j.ijcard.2020.08.062). These two aspects should be discussed.

Thank you for your important point of view. We added this paragraph to the limitation section:

“Furthermore, previous studies showed that Trp-Kyn metabolites are associated to a lesser extent with atrial fibrillation. The causality between Kyn and atrial fibrillation should be investigated in larger future studies. This is especially true in patients with HF, where atrial fibrillation is a very common co-morbidity.”

3) Some patients from the healthy control group suffer from comorbidities such as hypertension and AF. Even though this is not a relevant limitation, in the methods section it should be reported that control subjects were only free from heart failure.

Thank you for the valuable feedback. We added this paragraph to page number 4, line 12-13: “HC-subjects were free from HF. However, they had some risk factors such as arterial hypertension, atrial fibrillation, and diabetes mellitus.”

Minor points

  • Please put HC in brackets after healthy controls.

This was added in page 4 and in the abstract.

  • I see that in univariable and multivariable analyses prognostic factors other than biomarkers (i.e. atrial fibrillation, age, sex, etc.) were not tested. This limitation should be acknowledged.

Thank you for your remark. We added this paragraph to the limitation of our study on page 12: “Additionally, because of the small sample size, we tested in the multivariate analysis only a model consisting of different metabolites. However, testing other models including age, and other co-morbidites such as atrial fibrillation would be reasonable in larger cohorts.”

  • Please report the number of missing for each variable.

There is only one measurement missing of the endurance of left leg.

  • Please refer to latest available heart failure guidelines (doi: 10.1002/ejhf.2333.).

The new guidelines ESC-HF 2021 were cited.